# Mapping Panlongcheng: New Work on the Type-Site of the Early Shang Period (1500–1300 BC) in Hubei Province, China

Xin Su [1,2,†] , Qiushi Zou [2,3,*,†] , Shuchun Yao [4], Chunhai Li [4], Sanyuan Zhu [5] and Changping Zhang [2,3]

1    Department of Anthropology, Harvard University, Cambridge, MA 02138, USA; xsu@g.harvard.edu
2    Archaeological Institute for Yangtze Civilization, Wuhan University, Wuhan 430072, China; 00009171@whu.edu.cn
3    School of History, Wuhan University, Wuhan 430072, China
4    Nanjing Institute of Geography and Limnology, Chinese Academy of Sciences, Nanjing 210008, China; shchyao@niglas.ac.cn (S.Y.); chhli@niglas.ac.cn (C.L.)
5    Guangzhou Institute of Geochemistry, Chinese Academy of Sciences, Guangzhou 510640, China; zhusy@gig.ac.cn
*    Correspondence: zouqiushi@whu.edu.cn
†    These authors contributed equally to this work.

**Abstract:** Recent work at the early Shang period type site in Panlongcheng, Hubei Province, China, provides a new understanding of changes in the landscape and water environment over time. In the past few decades, the research at this site has obtained important results and shown progress in many aspects, but few scholars have discussed the geomorphological environment of Panlongcheng, especially the water environment. Researchers have long believed that the present-day environment and landscape of Panlongcheng are no different than during the early Shang period. However, recent archaeological discoveries indicate that there may still be some cultural remains underwater. Therefore, we used a combination of underwater surveys, drilling and digital mapping to expand our knowledge of the landscape of Panlongcheng during the early Shang period. This included mapping the lake basin using single-beam echo sounders and drilling to preliminarily observe the stratum and collect samples from underwater. We also conducted radiocarbon dating on the samples collected from the bottom of the lake. The results suggest that there might not have been a lake during the early Shang period. Therefore, the landscape and environment of Panlongcheng and other related issues should be reexamined. In addition, we hope the methods used in this study can provide a reference for related archaeological work in shallow water areas in inland China.

**Keywords:** early Shang period; underwater detection; landscape; water environment; GIS; radiocarbon dating

## 1. Introduction

As the most significant site in southern China (Figure 1) during the early Shang period (1500–1300 BC), Panlongcheng has attracted a great deal of attention from scholars since 1963. During more than 50 years of archaeological work, a larger number of cultural remains have been unearthed, and related detailed and in-depth studies based on these remains have been conducted, including studies of bronze [1–5], jade [6,7], ceramic [8,9] and stone artifacts [10]. In addition, there has been some progress in the areas of dating [11], cultural attributes [12–17], settlements [18] and ideology [19].

Many scholars believe that Panlongcheng may have been founded by the Shang people of the Central Plains and served as an outpost to acquire and control mineral resources because of the location of the site itself and the mineral resources in the surrounding area [15,20]. However, these assumptions and studies are inferred based on the present-day landscape and environment of the site. For example, the settlement is adjacent to a lake that connects to external rivers, forming a very convenient water transportation system,

which could not only link the surrounding small settlements and resources but could also maintain smooth and fast communication with the Central Plains.

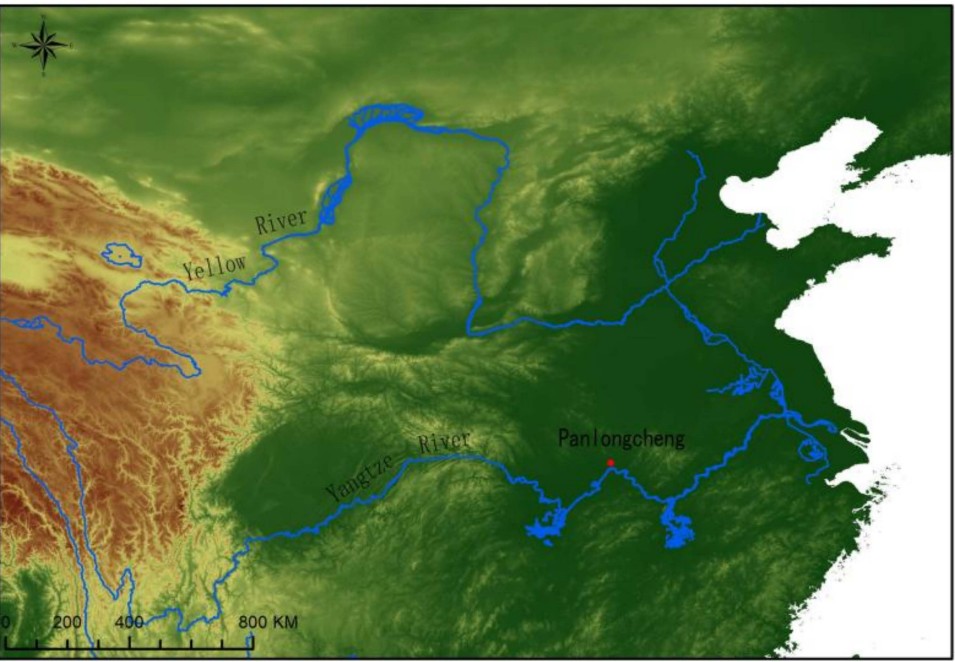

**Figure 1.** Map showing the location of the Panlongcheng site.

However, it seems to be problematic to use the contemporary landscape and environment as the premise of such a conclusion because it requires us to assume that the environment has not changed significantly over time. Obviously, for many sites that have existed for thousands of years, the environment in which they are located has often changed significantly. As was discovered at the Panlongcheng site in recent years, many ancient remains, such as burials and stone molds for casting bronzes, were exposed on the lake beach during the dry season [21,22]. In addition, some aerial photos and satellite images taken in recent decades have also shown that the waters around Panlongcheng have undergone significant changes (Figure 2). This suggests that there may be ancient remains under the current waters.

Therefore, from late 2016 to early 2017, we carried out underwater topographic mapping and underwater drilling in Panlong Lake at the Panlongcheng site. Through this series of work, we aimed to determine the appearance of the topography at the bottom of the Panlong Lake, the sedimentary situation at the bottom of the lake and when the lake formed. By acquiring this information about Panlong Lake, we hope to better understand the ancient environment and landscape of the Panlongcheng site and to provide new information and evidence for related research.

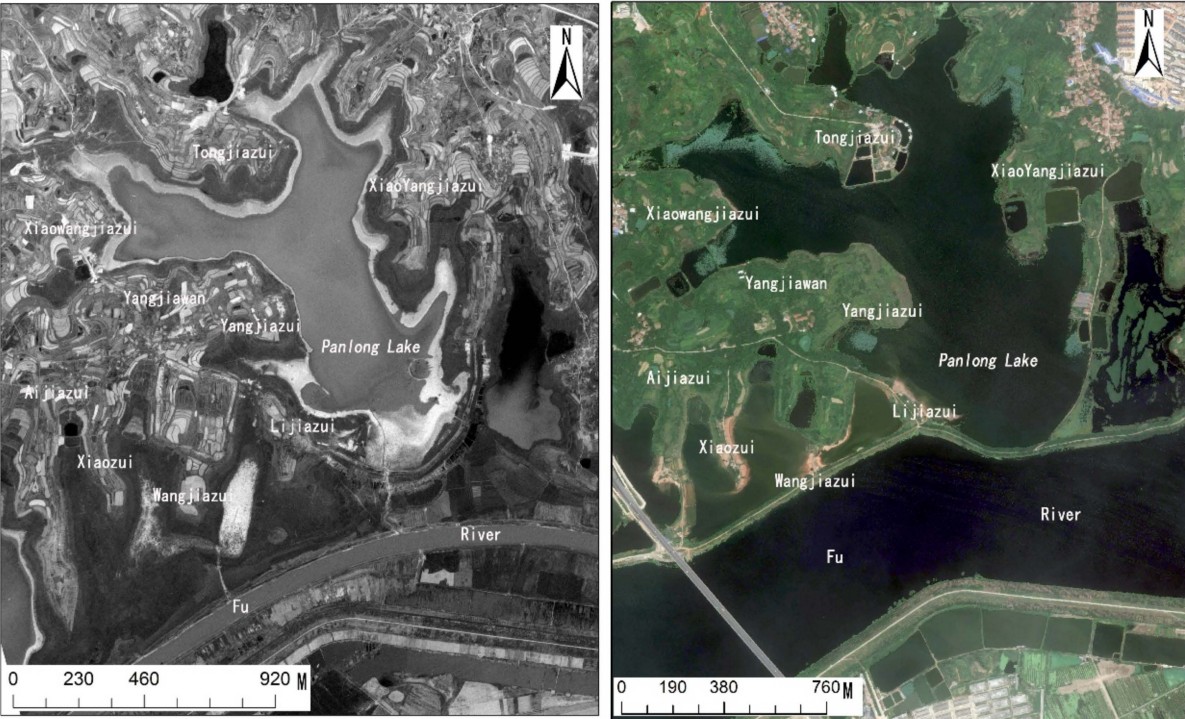

**Figure 2.** Satellite images of Panlongcheng (**Left** 1962; **Right** 2012).

## 2. Research Area and Context

Panlongcheng is a significant archaeological site associated with the Erligang culture, which is also known as the early Shang period (1500–1300 BC). This site is located on the northern bank of the Yangtze River and is surrounded by the Fu River and Panlong Lake in the current city of Wuhan, Hubei. Panlongcheng is currently the largest known early Shang period settlement or city in southern China, and it represents the southernmost reach of the Shang Dynasty at its peak. It was first discovered in 1954, and since 1963, archaeologists have been working on the site almost continuously.

According to previous studies of the Panlongcheng site, the site was sparsely inhabited during the Erlitou period (1900–1500 BC), mainly in the Wangjiazui in the southern part of the site. The approximate area of the entire settlement is about 0.2 km². During the early Shang period, the settlement expanded rapidly and dramatically, reaching an area of about 1 km². At the same time, the core of the settlement was moved from the south to the central area of the site, where a 75,000 m² town surrounded by rammed earth walls was founded. In addition, two large buildings were found in the northeastern corner of the town, and a patrician cemetery was discovered in the northeastern area outside of the town [15]. In the later stage of this period, the appearance of the entire settlement changed again. The core area was moved further north into the hills of Yangjiawan on the north side, while the central part of the town was abandoned. Since the elevation of the area where the Panlongcheng site is located gradually rises from south to north, archaeologists believe that the changes in the settlement's layout may have been due to the gradual rise of the water level, which forced people to migrate to higher places [18].

However, since the 1970s, archaeologists have repeatedly found early Shang period remains at the lakeside inside Panlongcheng. In particular, from 1980 to 1998, local archaeologists unearthed 14 Shang-period burials in the southeast corner of Yangjiazui (which is a part of Yangjiawan) when the lake water retreated [15,23]. From 2003 to 2013, archaeologists also uncovered an early Shang-period burial on the eastern side of Xiaozui [24] and collected several pieces of stone molds from the same period in the area nearby [21]. In addition, several early Shang period bronze and ceramic artifacts have also been found sporadically in the lakeside areas throughout the site.

If the rise of the water level caused the migration of people into the hills of Yangjiawan, why are some contemporaneous cultural remains found underwater? In addition to the explanation of the post-depositional process, these discoveries imply that the distribution of the early Shang-period remains in the Panlongcheng site may not only be limited to the current terrestrial area but may also extend to the area below the contemporary lake level.

## 3. Research Methods

### 3.1. Underwater Detection and Mapping

The underwater mapping of Panlong Lake was carried out in late 2016 and helped us initially explore the underwater topography. This work was conducted using single-beam echo sounders, which use a transducer to emit a short sound wave pulse vertically downwards. When the sound waves meet the bottom of the lake, they are reflected and return to the transducer. The depth of the water can be calculated from the two-way travel time of the sound waves and the average sound velocity within the water. Then, based on the current altitude of the lake surface and the detected water depth, we calculated the elevation of the lake bottom (Figure 3). The data collection process was to collect continuous points along the predetermined route [25–27]. The characteristics of data distribution are that the data along the route is very dense, and there is no data between these routes, so the data obtained using this method is linear rather than faceted. However, these data can be integrated through kriging to help clarify the situation of the entire lake bottom.

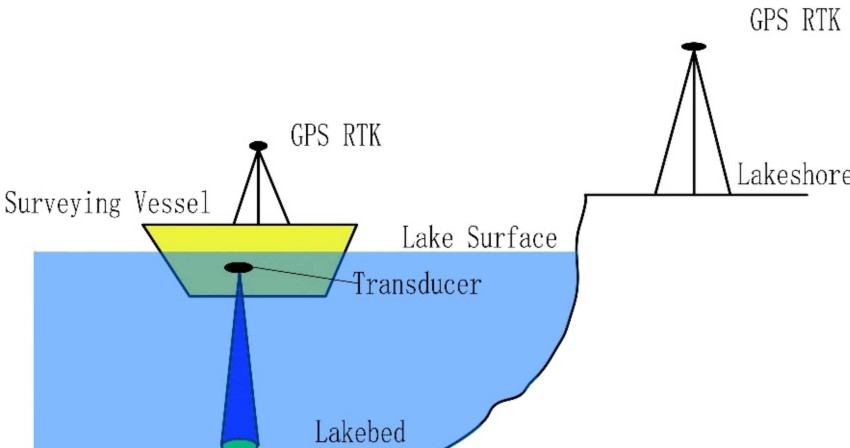

**Figure 3.** Schematic diagram of the working principle of the single-beam echo sounder.

In fact, most underwater archaeological mapping and detection projects currently being carried out in China use the multi-beam method rather than the single-beam method used in this study, such as studies of the shipwreck Dandong No.1 [28], the ancient city of Junzhou [29], the Shanglinhu kiln site [30] and the Jiangkou Chenyin site [31]. In these underwater archaeological studies, the multi-beam method has played a significant role in reconstructing underwater landscapes and detecting archaeological features beneath the water. The working principle of multi-beam waves is similar to that of single-beam waves, but due to the more complicated equipment, the multi-beam method can be used for surface detection instead of the linear detection of the single-beam method, which means that the multi-beam method has significant advantages in large-areas and fast data collection.

However, the cost of using a multi-beam system is relatively high, and because of its greater power and base array, it can often detect deeper and wider sections. Therefore, the multi-beam method is always used in some water areas, such as oceans [32,33], larger rivers and lakes [34–36], and huge reservoirs [29]. For small and shallow lakes and rivers in hinterlands, the maximum water depth is often only 5–6 m, the water level in the shoreside areas is less than 1 m, and the water area does not exceed 2 km$^2$ [37]. Therefore, it is not necessary to adopt the multi-beam method, especially when the budget is limited and the

workload is not too large. In addition, single-beam sonar, or side-scan sonar, has been shown to have good accuracy, low cost and effectiveness in several research projects and applications in recent years [27,38–41]. Therefore, since the area of Panlong Lake is about 1 km², its depth is about 0–4 m, and our funding was limited, the single-beam method was determined to be more suitable in this case.

In the survey, we used a boat equipped with a Hi-Target H32 Real-Time Kinematic (RTK) Global Positioning System (GPS) and an HD-MAX single-beam sounder to detect the underwater topography of the Panlong Lake. The single-beam echo sounder has an accuracy of ±1 cm and a sounding range of 0.15–300 m. Considering the draught of the boat (0.9 m), this work could not carry out surveying and mapping for water areas with depths of less than 1 m. The specific work was divided into the following two parts.

First, the data were collected in the field. A reference system for surveying and mapping was established at Panlongcheng, and several permanent control points were set up as well, directly providing a spatial reference for underwater detection. After the data were collected, survey lines were laid out at 25 m intervals and the depth was measured every 7 m. The survey ship sailed along the arranged route (Figure 4). The sounder performed the data collection and storage with the cooperation of the RTK GPS. After the preliminary calibration of the data, the determination of the topography of the lake bottom was completed.

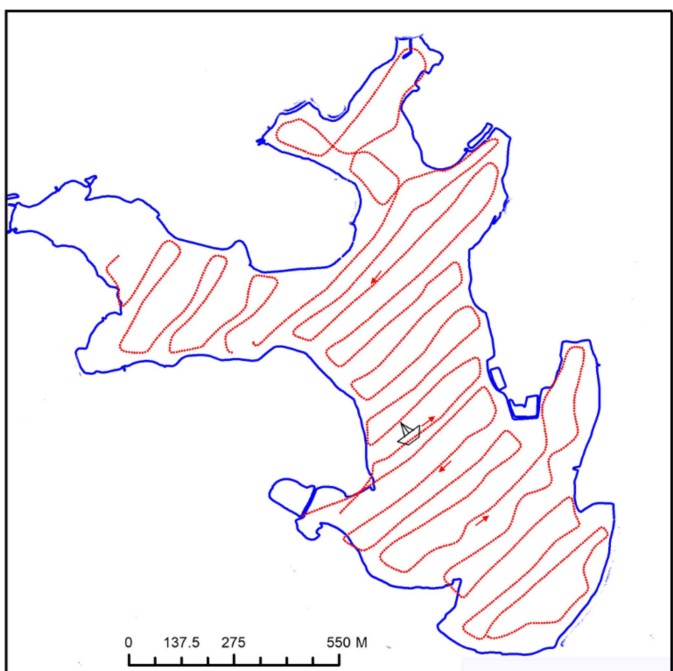

**Figure 4.** Map showing the trajectory of the survey ship. Each red dot indicates a measurement point.

Second, the data were processed in the lab. AutoCAD and ArcGIS were used to make a digital elevation model (DEM) of the entire lake area. First, we jointly edited the existing 1:2000 topographic map of Panlongcheng and the underwater data to draw a 1:2000 digital line graph (DLG), which covered the entire site, including the areas above water and underwater. Then, based on the generated 1:2000 DLG, a triangular irregular network (TIN) was produced in ArcGIS. Finally, we transformed the TIN into a DEM in raster mode to complete the data processing (Figure 5).

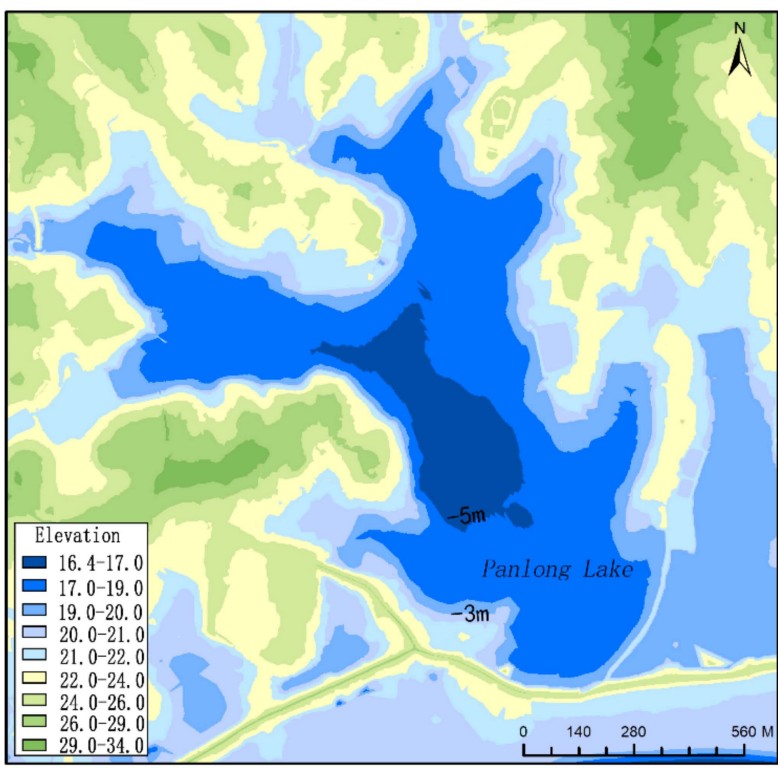

**Figure 5.** Topographic map of Panlong Lake obtained after the survey and data processing.

### 3.2. Underwater Drilling and Sample Collection

After the mapping, we conducted the first underwater drilling using an underwater exploration platform. The goals of this work were to clarify the distribution of the cultural remains underwater and to understand the deposition in the lake basin. The underwater exploration platform was mainly composed of a rubber airbag, carbon fiberboard, aluminum alloy drill frame and drilling tools (Figure 6). The drilling tools included a drill pipe and sampler. The samplers selected for this work can be divided into two types: semi-circular samplers and piston samplers. The semi-circular sampler is similar in shape to an archaeological shovel called a Luo yang Shovel, and its advantage is that the basic conditions in the sampler can be directly observed on site. The piston sampler has a built-in PVC sampling tube with a length of 1.5 m and an inner diameter of 5.9 cm, which can seal the sample in the PVC tube during the sampling process so that the sample can be transported back to the laboratory for related testing and analysis.

Considering the limited resources, work efficiency, time and other related issues, we were unable to conduct large-scale intensive drilling of the entire lake. Therefore, we adopted the following method. First, we set up drilling holes along the lake bank with an interval of 50 m between each hole in order to explore whether the cultural remains on the land extended underwater. If cultural remains were found at a sampling location, the area was drilled intensively. Second, in the offshore area, we also set up three lines for drilling to gain a more comprehensive understanding of the deposits at the bottom of the lake (Figure 7). Similarly, the drilling points on each line were 50 m apart.

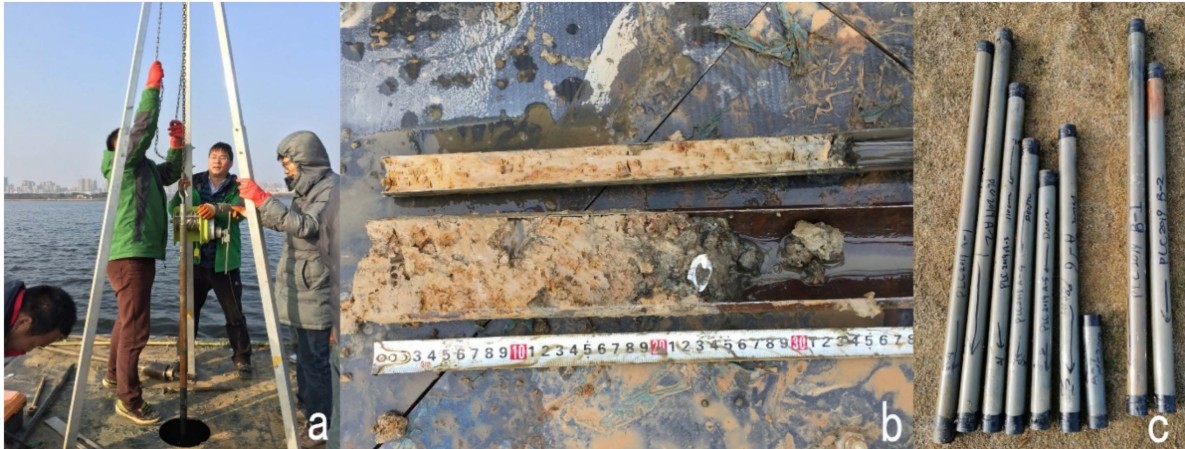

**Figure 6.** (**a**) Drilling work in progress and using the piston sampler to collect samples from the lake bottom. (**b**) The holes containing cultural remains from the early Shang period. (**c**) The samples sealed in PVC tubes.

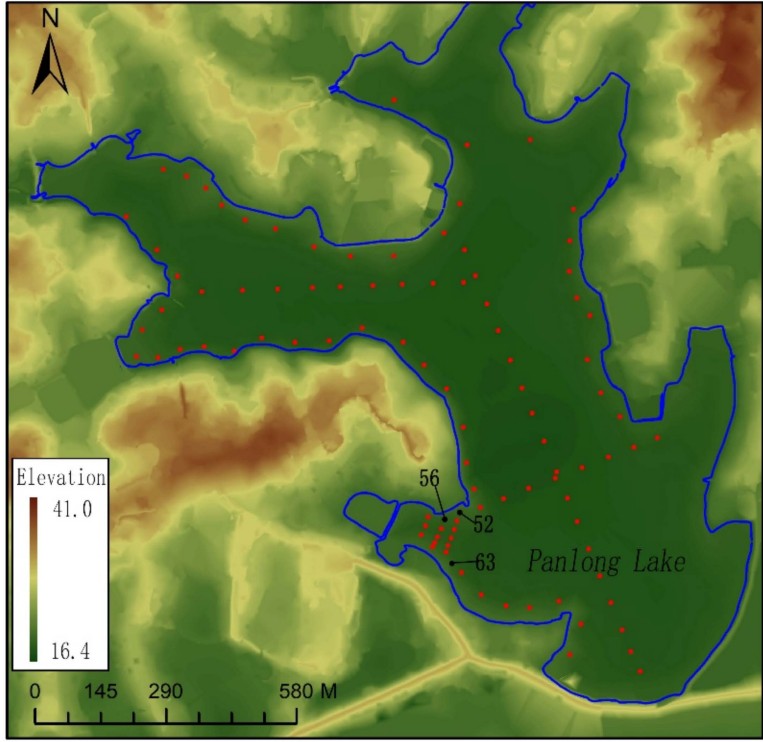

**Figure 7.** The distribution of the underwater drilling holes, in which some pottery sherds were found in No. 52, 56 and 63 drilling holes.

### 3.3. Radiocarbon Dating

The preliminary dating of the sediments from the lake bottom was one of our important goals, so we conducted radiocarbon dating on organic samples collected during drilling.

The pretreatment of the samples and the Accelerator Mass Spectrometry (AMS_-$^{14}$C analysis were carried out in the $^{14}$C Accelerator Mass Spectrometry Laboratory of the Guangzhou Institute of Geochemistry, Chinese Academy of Sciences. The dating procedure is summarized as follows. The samples of plant materials (~100 mg) were mechanically cleaned with distilled water in an ultrasonic bath. Next, the samples were chemically pretreated as follows: (1) 1 M HCl at 80°C for 3 hr (repeated four times); and (2) distilled water after each stage and for final rinsing (repeated until pH=7). After freezing and drying, 6–7 mg of each prepared sample was combusted in the presence of granular CuO and

wire Ag in an evacuated, flame-sealed glass container for 3 hr at 900°C. The resultant $CO_2$, having been purified through successive application of liquid nitrogen, ethanol and dry ice, was reduced to graphite with twice the amount of $H_2$ as the reductant for 2–3 hr at 525°C. Finally, the graphite samples were pressed into aluminum holders for the AMS analysis.

## 4. Results

According to the acoustic data, the bottom of Panlong Lake is relatively flat, which is similar to other lakes located in the Jianghan area [37], and the elevation of the deepest part is about 16.4 m above sea level. Given that the elevation of the lake surface is 21–22 m above sea level (because of seasonal fluctuations), the deepest part of the lake reaches about 5 m.

Through underwater drilling, we found that there are at least two layers of deposits at the bottom of the lake. The first layer is about 0.3–2.1 m thick and consists of gray-brown silt, and it is typical lacustrine sediment containing abundant aquatic animal and plant residues. In addition, some modern remains were also found in this layer, so we speculate that this layer was formed in modern times.

The second layer is dense and hard and consists of blue-gray hard clay, and we could not drill through it, so its thickness is unknown. The sediments of this layer are stiff clay, and no aquatic animal and plant residues and micro-fossils were found, so this layer should be continental sediments. In addition, some pottery sherds from the early Shang period were found in this layer at a depth of 0.5–1.1 m below the lake floor (Figure 8), and no remains from other periods were discovered.

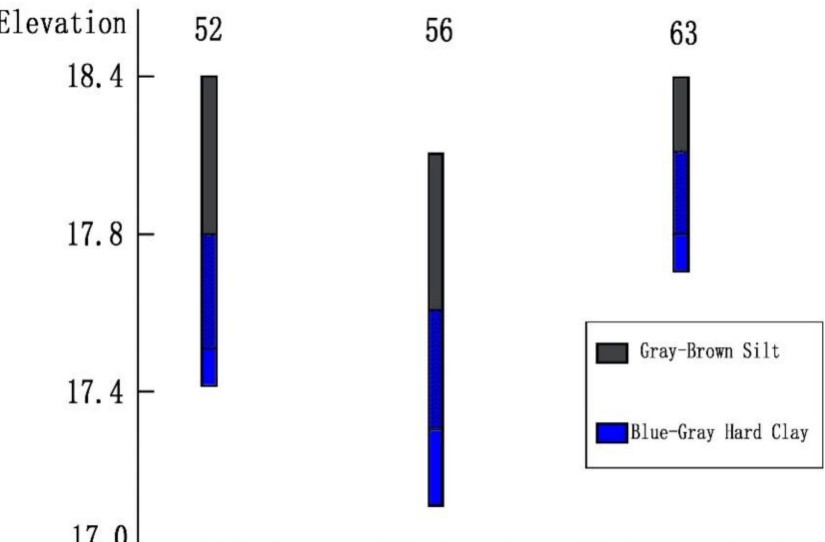

**Figure 8.** The sediment situation of No. 52, 56 and 63 drilling holes. In which, the black spots in the layer 2 (blue-gray hard clay) indicate the depth range where pottery sherds were found.

During the processing of the drill cores, we obtained a total of 6 plant residues in the second layer of sediments at the bottom of the lake, of which two samples were from in sediments at depths of 2.6–3 m below the lake floor, and the other four were found at depths of 5.8–6.3 m below the lake floor. After processing the samples, only the four from depths of 5.8–6.3 m were available. The results show that the samples date to 890–600 BC (Table 1).

**Table 1.** Calibrated radiocarbon dates from the second layer of sediments at the bottom of Panlong Lake. All of the dates were provided by the [14]C Accelerator Mass Spectrometry Laboratory of the Guangzhou Institute of Geochemistry, Chinese Academy of Sciences.

| Lab Number | Sample | Carbon 14 Date (BP) | 1σ (68.3%) | 2σ (95.4%) |
|---|---|---|---|---|
| GIG-UCIB0030 | Plant Fibers | 2465 ± 25 | 763 (68.3%) 691 BC | 775 (90.0%) 607 BC<br>595 (5.5%) 538 BC |
| GIG-UCIB0031 | Plant Fibers | 2565 ± 25 | 794 (68.3%) 770 BC | 806 (91.6%) 750 BC<br>683 (2.7%) 669 BC<br>607 (1.1%) 597 BC |
| GIG-UCIB0032 | Plant Fibers | 2650 ± 25 | 818 (68.3%) 797 BC | 888 (0.9%) 882 BC<br>835 (94.6%) 786 BC |
| GIG-UCIB0033 | Plant Fibers | 2635 ± 25 | 812 (68.3%) 793 BC | 825 (95.4%) 781 BC |

A single piece of data often only reflects its own time point and cannot well reflect the time span of an event. In this case, although we do not have any samples from the bottom of the second layer and cannot know when the second layer was formed, we do have a group of data from depths of 5.8–6.3 m below the lake floor, which means that it is possible to estimate the formation period of the sediments at this depth. Therefore, by running the grouping and boundary model in OxCal v4.4.4, we determined that the sediments at depths of 5.8–6.3 m below the lake floor were deposited beginning at about 1109–734 BC (95.4%) and ending at 803–400 BC (95.4%) (Figure 9). If the formation time at this depth is no earlier than 1109 BC, then the formation time of the second layer should be no later than this time. If the end of the sediment deposition at this depth occurred no later than 400 BC, then there was still land in this area at least at 400 BC. Therefore, in the early Shang period (1500–1300 BC), Panlongcheng may have been located on relatively open land instead of being located near a lake.

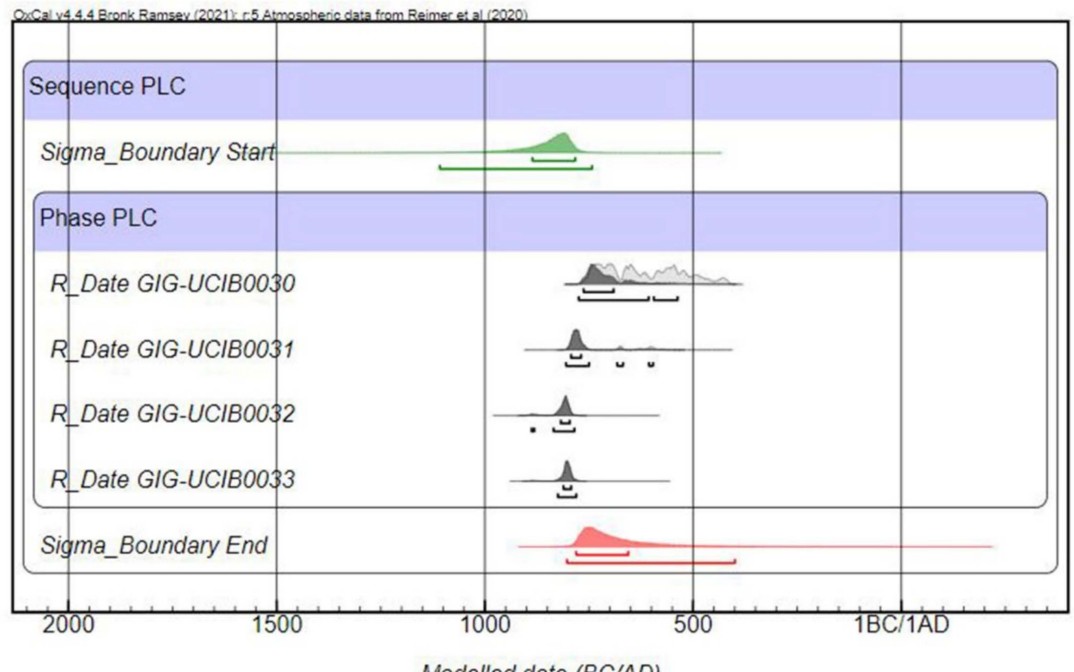

**Figure 9.** *Cont.*

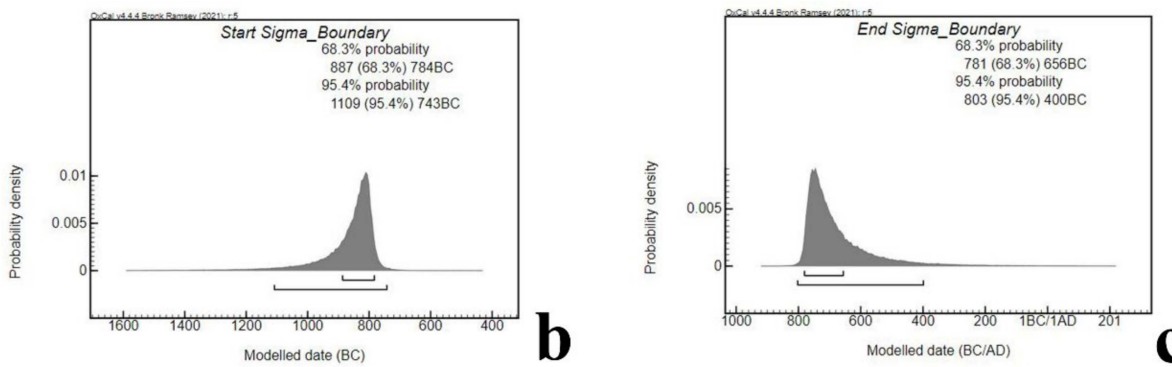

**Figure 9.** An illustration showing the summary dates as well as modeled start and end dates. (**a**). the summary of dates, including the start, end and all collected samples. (**b**). the calibration of start date (**c**). the calibration of the end date.

## 5. Discussion

### 5.1. Ancient Landscape, Environment and Related Issues

The hydrological environment is very important for the Panlongcheng site for two reasons. Previous studies based on the environment of Panlongcheng concluded that the ideal river and lake conditions around the site were an important reason why the Shang people from the Central Plains established a large-scale settlement here. In addition, the changes in the entire settlement scheme also seem to be closely related to the rise of the water level of Panlongcheng Lake. Even though these inferences seem very reasonable from the perspective of the contemporary landscape and environment, whether these inferences are still tenable needs to be reconsidered if the landscape and environment around the site have undergone significant changes in the past millennia. Interestingly, based on several archaeological discoveries in recent years, increasing evidence indicates that there are some cultural remains underwater. This may imply that the area currently underwater may have been dry land at that time.

Under this assumption, we carried out survey and drilling work in Panlong Lake. Relying on new technologies and methods, the situation at the bottom of the lake was gradually clarified. First, the bottom of Panlong Lake is relatively flat, and the cultural remains are still under the modern layer at the bottom of the lake today. Second, the layer below the modern sediments is continental sediments, and through dating, it was determined that the deposition of these continental sediments began no later than 1100 BC and the deposition ended no earlier than 400 BC. Considering the major phase of Panlongcheng is the early Shang period (1500–1300 BC), the lake we see today may not have formed yet when people lived in Panlongcheng. Instead, a large area of low-lying, flat land was likely the major landscape and environment at that time.

The inferences based on the contemporary landscape and environment are problematic. The Shang people from the Central Plains may have established a large settlement here to gain control of the surrounding ore resources, but whether convenient water transportation conditions were also an important factor needs to be reconsidered. In addition, since the formation of the lake occurred after the abandonment of Panlongcheng, the continuous shift of the settlement's center to higher ground would not have been due to the rise of the water level of the lake. Therefore, factors beyond the environment, such as geographic conflicts or strategic adjustments, should be considered when interpreting this phenomenon.

In addition, our discoveries underwater may provide new clues and research possibilities for aspects that we did not previously know. For such a large settlement, previous studies on the subsistence economy and residential area are very rare. This is because no large-scale residential areas have been found and unearthed in previous archaeological studies. In addition, the acidic soil environment in southern China has caused a lot of the organic materials to decompose, making it difficult to find and collect samples during excavations. However, in this underwater mapping and drilling, we found that the bottom

of the lake is relatively flat, from which it can be inferred that the general residential area and crop planting area may be under the lake. Even though there is no clear, direct evidence to prove these hypotheses, our findings provide crucial clues for investigating these issues.

*5.2. Method*

The research methods used in this study mainly include the use of single-beam echo sounders to survey and map the underwater terrain and the use of underwater exploration platforms to drill and sample the lake basin area. Even though both single-beam and multi-beam echo sounders are ultrasonic sounding systems, in recent years, only the multi-beam method has been widely used for topographic surveying and underwater archaeology work in water, such as offshore waters and large and deep inland lakes in China. The Panlongcheng site studied in this article is located in the middle reaches of the Yangtze River. The lakes in this area are generally characterized by tortuous lake banks, shallow depths, and deep silt deposits at the bottom of the lakes. Compared with multi-beam echo sounders, single-beam echo sounders are still powerful and are less expensive. Therefore, the single-beam echo sounder is more suitable for working in the relatively shallow, small inland lakes in the middle reaches of the Yangtze River.

The underwater exploration platform was originally designed as special equipment used by environmentalists to sample lake sediments, not for archaeologists in archaeological work. However, the underwater archaeological drilling at Panlong Lake demonstrated that the platform has a good working performance. For lakes similar to Panlong Lake with shallow water depths and deep silt deposits at the bottom of the lake, ancient cultural relics are often covered by several meters of modern silt layers. Therefore, searching for ancient cultural deposits at the bottom of the lake using diving and other methods may not achieve the desired results, and the cost of underwater archaeological methods such as diving is much higher than that of underwater drilling. Therefore, drilling and sampling at the bottom of shallow inland lakes using underwater exploration platforms can provide direct information about the situation of the bottom of the lake. In addition, the collected samples can also provide a good foundation for other work, including radiocarbon dating, pollen and phytolith analysis. This method is very effective for exploring ancient remains distributed in inland lakes.

For a long time, underwater archaeological work in China has mostly been carried out in large offshore and deep inland reservoirs, with the main purpose of exploring ancient shipwrecks or large-scale underwater cities or settlements. However, in the middle and lower reaches of the Yangtze River, the rivers and lakes are ubiquitous environments. Under such geographic conditions, ancient sites are mostly distributed along rivers or lakes. Therefore, many sites are located not only on land but also extending into modern water bodies. In the past, we have not been able to carry out effective exploration of the ancient remains distributed in rivers and lakes, so we lack a comprehensive understanding of the geographic environment and layout of ancient settlements. The underwater archaeological work methods used at the Panlongcheng site in this study provide us with new ideas and methods for underwater archaeological work in inland lakes.

## 6. Conclusions

Through underwater survey and drilling, the cultural remains of the early Shang period under the modern silt layer of the lake basin have been found. These remains were found at an altitude of about 17.5 m, which is significantly lower than the lake water level at present. At the same time, the radiocarbon dating results also show that the formation time of the layer under the modern silt at the bottom of the lake should not be earlier than 1100 BC. All the evidence indicates that the Panlong Lake may not yet be formed during the main use phase of Panlongcheng, which means that Panlongcheng at that time should be based on the terrestrial landscape and environment instead of what we see today. Accordingly, some previous understandings based on the contemporary landscape and environment of Panlongcheng may need to be reconsidered.

In terms of method, the research methods used in this study provide a good reference for the exploration of underwater archaeological remains in inland lakes in China. Compared with the use of large vessels to salvage sunken ships, and the collection and recording of underwater relics through diving, the equipment used in this study is relatively simple, and the difficulty and cost of the work are lower. However, in terms of reconstructing the geographic environment and overall layout of ancient settlements, it was demonstrated to be an effective method that other methods cannot replace.

The landscape and environment are not static. As time passes, they change. The goal of archaeological research is to try to determine what happened in the past. Through a series of methods, archaeologists explore and interpret the past. When we study a site, we are confronted with an object that is in the present time and space and has been abandoned, and it is difficult to determine the situation of the site when it was in use and the environment in which it was located. Under such conditions, the interpretation of the past is obviously biased. Therefore, it is necessary to reconstruct the original landscape and environment of the area in which the site was located before conducting in-depth research and interpretation.

**Author Contributions:** Conceptualization: X.S. and Q.Z.; methods: X.S.; software: Q.Z. and S.Z.; validation: X.S. and Q.Z.; investigation: X.S., Q.Z., S.Y. and C.L.; data curation: S.Z.; writing—original draft preparation: X.S. and Q.Z.; writing—review and editing: X.S. and Q.Z.; visualization: X.S. and Q.Z.; project administration: C.Z. All authors have read and agreed to the published version of the manuscript.

**Funding:** This research was funded by the National Social Science Fund of China (Major Project Grant# 16ZDA146) and the China Postdoctoral Science Foundation (#2020M682463).

**Data Availability Statement:** The data is contained within this article.

**Acknowledgments:** We are grateful to the following individuals for their contributions to the project: Qingwu Hu, Rowan Flad, Yinghua Li and Yuduan Zhou. We are also grateful to the many graduate students who joined us in the field or otherwise contributed to this work: Hang Liao, Shen Xu, Jindong Lu, Dongnian Duan, Dechuan He and Chaohao Lin. We want to thank the following institutions for their generous support during our fieldwork: The School of History at Wuhan University, The Archaeological Institute of Hubei Province, The Panlongcheng Site Museum, The Nanjing Institute of Geography and Limnology and The Guangzhou Institute of Geochemistry, Chinese Academy of Sciences. Finally, we thank the people of Panlongcheng for their support of our work in the field and their constant generosity year after year.

**Conflicts of Interest:** The authors declare that there are no conflicts of interest.

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
