# Peer review of "Mapping Panlongcheng: New Work on the Type-Site of the Early Shang Period (1500–1300 BC) in Hubei Province, China"

_land, doi:10.3390/land10101033_

Round 1
Reviewer 1 Report
- It would be convenient to propose some hypothesis about the causes of the origin of the lake (anthropic, natural, climatic, etc.).
- Is any comparative analysis with nearby and similar cases possible?
- Some bibliographical reference on research lines dedicated to the historical interaction society-environment could be of interest.
Author Response
We have addressed each of the comments and questions, and we believe the revised manuscript is now stronger and clearer. In a separate document, we provide a point-by-point response to each question, as requested.

Reviewer 2 Report
The manuscript presented needs more work to become acceptable for publication in my mind. In general, the manuscript will greatly benefit by a grammar check by a native English speaker. As the text is now it is sometimes difficult to understand what exactly the authors are trying to say.
Specific comments
Introduction
At the end of the introduction (line 61-67) some results of the study is presented. This part should be moved (or removed) to the results section as the introduction section should not have results from the current study. Instead the end of the introduction should present the aim and some clear research questions for your study, which is lacking at present.
Line 61-63 – it is not described what kind of cultural remains you have found.
Line 65 – C14 data from what material?
Figure 2 – It would be very helpful if the excavation area/Panlongcheng area is shown in the figure.
Research area and context
Line 79-93 – A figure showing a map over the area with the places mentioned in the text is needed to understand the study better. For example the places Yangjiazui, Yangjiawan and Xiaozuo and the Panlongcheng area.
The paragraph 79-93 lacks references to some sentences.
Research methods
Figure 5 – the figure needs a legend describing the height/depth of the green and blue colors.
The very extensive description of the C14 methodology in chapter 3.3 is not needed. Instead I would prefer a description of your samples. How many samples? What kind of material was sampled? What was the sample depth?
Figure 7 – the figure lacks a legend to describe the color changes. It would also good to mark the extent of the lake in the figure.
Results
The results are difficult for the reader to understand. The important layer 2 is not described more than it was hard and therefore difficult to core and that it is of an unknown thickness. We do not know if the material is aquatic or terrestrial which is of great importance for the interpretation. We also do not know where in layer 2 the C14 samples come from (depth etc.) which we need to know in order to interpret the age of layer 2.
What is needed is a description of the sediment sequence, and preferably a new figure with a profile of the drillings so that we can see the thickness of layer 1 and 2 over distance.
Another thing which is not clear nor systematic is the way to describe elevation. Figure 5 suggest that the lake is not more than 6 meter deep, but in the text (line 252) it is stated that the deepest part of the lake is 16.4 m!
Normally the depth below the lake floor is used for describing the depth of sediment sequences and samples found in the drillings. It is also mentioned that some remains were found at an elevation of 17.5 m (line 261). Is that m above sea level? You also write on line 261-262 that the remains are found below the current lake surface, which is obvious as you have drilled below the lake bottom to find the remains.
Line 260 – It is not possible to locate the holes on figure 7. It might be good to put those numbers (52, 56 and 63) in the figure for better understanding.
The C14 samples are from layer 2 but it is not stated how thick layer 2 is or from where in the layer the samples are obtained from. We also do not know the depth relation between the four samples. Yet the paper states the maximum and minimum ages of the layer 2, 1109 BC to 400 BC. That is not possible, as the C14 samples are clearly not from the bottom of layer 2 (the drilling didn’t reach the bottom of the layer) so then the oldest age of layer 2 is still unknown. All you can say with the data presented in the paper is that the C14 dated remains indicate that the sampled part of layer 2 was deposited during ca 800-700 BC.
Discussion
In the discussion the authors say that the lake was probably not formed during the time of Panlongcheng, but the data to make that claim is not presented in the paper. The reader do not know whether layer 2 is made up of terrestrial or aquatic sediments. We also do not know the age extent of layer 2, as discussed above, and therefore we do not know the age of the Panlong Lake.
The authors must present more data to support their interpretations and conclusion in order to accept the manuscript. It might be that your interpretation is correct, but all available data to support it needs to be presented to the reader.
Furthermore, improving the figures according to my suggestions would greatly benefit the manuscript.
Author Response

(The authors gave the same response as above.)

Reviewer 3 Report
The paper touches upon a very interesting point in the archaeology of the Bronze Age - the Panlong City site in Hubei. This topic has seeт intense coverage in academic literature, particularly Chinese. The presented paper covers the implication of underwater archaeology for investigating Panlong Lake, which due to rising water levels flooded areas of the archaeological site (60).
The authors have used two field data collecting methods - sonar mapping and benthic core collecting. A singlebeam sonar had been used for constructing the bathymetry of Panlong Lake. While, multibeam sonar is, indeed, a more effective tool for mapping underwater environments than singlebeam devices, I do not suppose there is a need for a detailed discussion of why the authors had used latter. Modern multibeam systems can be very compact and mounted on most watercraft, especially if the researchers’ boat drew a meter of water (154). Therefore, a simple explanation of low funding would have been sufficient to explain the choice of a singlebeam device. On the other hand, for mapping the study area where, as the authors indicate, depths do not exceed 4 m (149), an inexpensive recreational sonar would suffice. These devices are often coupled with a sidescan sonar making them ideal for shallow water mapping and imaging (see: Kaeser, 2013; Buscombe, 2017). The authors might like to consider using a similar device for future investigations. Sidescan sonar imagery of the lakebed would also be contribute to the investigation. I suggest that the discussion on multibeam sonars should be replaced with a more comprehensive and appropriate review of sonar usage for lake, riverine, and shallow water archaeology.
The benthic core sample investigation method, described in the paper is interesting and might be considered novel for limnological environments in the studied part of the world. The fact that the research team we able to investigate the harvested cores using carbon dating adds to the value of the paper. However, the detailed description of the laboratory testing procedures (218-248) should be shortened. A cross-section image of stratigraphic data would benefit the paper and assist interpreting the results of the investigation.
Furthermore, I believe that the results of the fieldwork might be expanded for the reader to be able to get a clearer understanding of how core sampling contributes to Shang archaeology.
While comprehensible, the text requires some serious language vetting, especially in relation to tense usage and sentence structure. Besides, the text is dotted with occasional colloquialisms (e.g.: a lot (34, 56); lab (169)) that should be avoided. The authors might also reconsider article usage and their choice of terminology.
I do recommend the paper for publication after editing. Perhaps the authors could find an Anglophone speaker to check their English.
Suggested literature:
- Daniel Buscombe, Shallow water benthic imaging and substrate characterization using recreational-grade sidescan-sonar, Environmental Modelling & Software, Volume 89, 2017, Pages 1-18, https://doi.org/10.1016/j.envsoft.2016.12.003.
- Kaeser, Adam J., Thomas L. Litts, and T. Wesley Tracy. "Using low‐cost side‐scan sonar for benthic mapping throughout the lower Flint River, Georgia, USA." River Research and Applications 29.5 (2013): 634-644.
- Blondel, Philippe. The handbook of sidescan sonar. Springer Science & Business Media, 2010.
- Handley, Sean J., et al. "Historical analyses of coastal marine sediments reveal land-based impacts on the benthos." New Zealand Journal of Ecology 44.2 (2020): 1-10.
- Westley, Kieran, and Rory Mcneary. "Archaeological Applications of Low‐Cost Integrated Sidescan Sonar/Single‐Beam Echosounder Systems in Irish Inland Waterways." Archaeological Prospection 24.1 (2017): 37-57.
Author Response

(The authors gave the same response as above.)

Round 2
Reviewer 2 Report
The authors have satisfactory reacted to my concerns to the manuscript and it is now acceptable for publication.